# Transfer Learning in a Transductive Setting

**Marcus Rohrbach**      **Sandra Ebert**      **Bernt Schiele**

Max Planck Institute for Informatics, Saarbrücken, Germany
`{rohrbach,ebert,schiele}@mpi-inf.mpg.de`

## Abstract

Category models for objects or activities typically rely on supervised learning requiring sufficiently large training sets. Transferring knowledge from known categories to novel classes with no or only a few labels is far less researched even though it is a common scenario. In this work, we extend transfer learning with semi-supervised learning to exploit unlabeled instances of (novel) categories with no or only a few labeled instances. Our proposed approach *Propagated Semantic Transfer* combines three techniques. First, we transfer information from known to novel categories by incorporating external knowledge, such as linguistic or expert-specified information, e.g., by a mid-level layer of semantic attributes. Second, we exploit the manifold structure of novel classes. More specifically we adapt a graph-based learning algorithm – so far only used for semi-supervised learning – to zero-shot and few-shot learning. Third, we improve the local neighborhood in such graph structures by replacing the raw feature-based representation with a mid-level object- or attribute-based representation. We evaluate our approach on three challenging datasets in two different applications, namely on *Animals with Attributes* and *ImageNet* for image classification and on *MPII Composites* for activity recognition. Our approach consistently outperforms state-of-the-art transfer and semi-supervised approaches on all datasets.

## 1  Introduction

While supervised training is an integral part of building visual, textual, or multi-modal category models, more recently, knowledge transfer between categories has been recognized as an important ingredient to scale to a large number of categories as well as to enable fine-grained categorization. This development reflects the psychological point of view that humans are able to generalize to novel[1] categories with only a few training samples [17, 1]. This has recently gained increased interest in the computer vision and machine learning literature, which look at zero-shot recognition (with no training instances for a class) [11, 19, 9, 22, 16], and one- or few-shot recognition [29, 1, 21]. Knowledge transfer is particularly beneficial when scaling to large numbers of classes [23, 16], distinguishing fine-grained categories [6], or analyzing compositional activities in videos [9, 22].

Recognizing categories with no or only few labeled training instances is challenging. To improve existing transfer learning approaches, we exploit several sources of information. Our approach allows using (1) trained category models, (2) external knowledge, (3) instance similarity, and (4) labeled instances of the novel classes if available. More specifically we learn category or attribute models based on labeled training data for known categories $y$ (see also Figure 1) using supervised training. These trained models are then associated with the novel categories $z$ using, e.g. expert or automatically mined semantic relatedness (cyan lines in Figure 1). Similar to unsupervised learning [32, 28] our approach exploits similarities in the data space via a graph structure to discover dense regions that are associated with coherent categories or concepts (orange graph structure in Figure 1). However, rather than using the raw input space, we map our data into a semantic output space with the

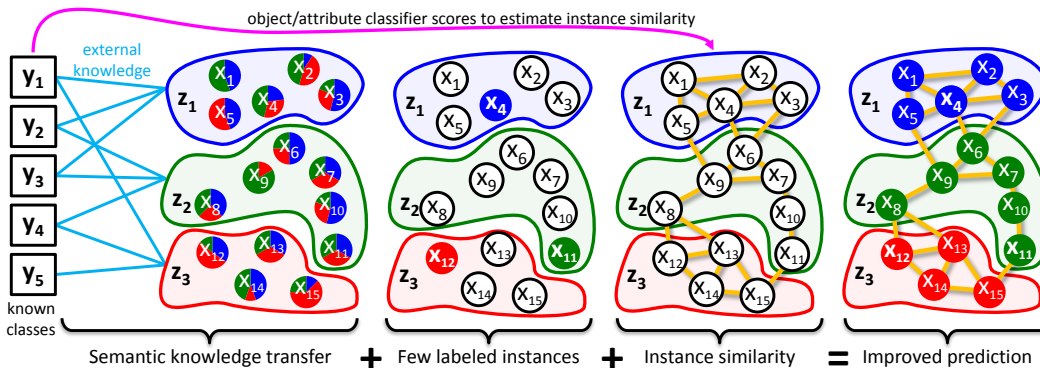

Figure 1: Conceptual visualisation of our approach Propagated Semantic Transfer. Known categories $y$, novel categories $z$, instances $x$ (colors denote predicted category affiliation). Qualitative results can be found in supplemental material and on our website.

models trained on the known classes (pink arrow) to benefit from their discriminative knowledge. Given the uncertain predictions and the graph structure we adapt semi-supervised label propagation [34, 33] to generate more reliable predictions. If labeled instances are available they can be seamlessly added. Note, attribute or category models do not have to be retrained if novel classes are added which is an important aspect e.g. in a robotic scenario.

The main contribution of this work is threefold. First, we propose a novel approach that extends semantic knowledge transfer to the transductive setting, exploiting similarities in the unlabeled data distribution. The approach allows to do zero-shot recognition but also smoothly integrate labels for novel classes (Section 3). Second, we improve the local neighborhood structure in the raw feature space by mapping the data into a low dimensional semantic output space using the trained attribute and category models. Third, we validate our approach on three challenging datasets for two different applications, namely on *Animals with Attributes* and *ImageNet* for image classification and on *MPII Composites* for activity recognition (Section 4). We also provide a discussion of related work (Section 2) and conclusions for future work (Section 5). The implementation for our Propagated Semantic Transfer and code to easily reproduce the results in this paper is available on our website.

## 2  Related work

Knowledge transfer or transfer learning has the goal to transfer information of learned models to changing or unknown data distributions while reducing the need and effort to collect new training labels. It refers to a variety of tasks, including domain adaptation [25] or sharing of knowledge and representations [30, 3] (a recent categorization can be found in [20]).

In this work we focus on transferring knowledge from known categories with sufficient training instances to novel categories with limited training data. In computer vision or machine learning literature this setting is normally referred to as zero-shot learning [11, 19, 24, 9, 16] if there are no instances for the test classes available and one- or few-shot learning [16, 9, 8] if there are one or few instances available for the novel classes.

To recognize novel categories zero-shot recognition uses additional information, typically in the form of an intermediate attribute representation [11, 9], direct similarity [24] between categories, or hierarchical structures of categories [35]. The information can either be manually specified [11, 9] or mined automatically from knowledge bases [24, 22]. Our approach builds on these works by using a semantic knowledge transfer approach as the first step. If one or a few training examples are available, these are typically used to select or adapt known models [1, 9, 26]. In contrast to related work, our approach uses the above mentioned semantic knowledge transfer also when few training examples are available to reduce the dependency on the quality of the samples. Also, we still use the labeled examples to propagate information.

Additionally, we exploit the neighborhood structure of the unlabeled instances to improve recognition for zero- and few-shot recognition. This is in contrast to previous works with the exception of

the zero-shot approach of [9] that learns a discriminative, latent attribute representation and applies self-training on the unseen categories. While conceptually similar, the approach is different to ours, as we explicitly use the local neighborhood structure of the unlabeled instances. A popular choice to integrate local neighborhood structure of the data are graph-based methods. These have been used to discover a grouping by spectral clustering [18, 14], and to enable semi-supervised learning [34, 33]. Our setting is similar to the semi-supervised setting. To transfer labels from labeled to unlabeled data *label propagation* is widely used [34, 33] and has shown to work successfully in several applications [13, 7]. In this work, we extend transfer learning by considering the neighborhood structure of the novel classes. For this we adapt the well-known label propagation approach of [33]. We build a k-nearest neighbor graph to capture the underlying manifold structure as it has shown to provide the most robust structure [15]. Nevertheless, the quality of the graph structure is key to success of graph-based methods and strongly dependents on the feature representation [5]. We thus improve the graph structure by replacing the noisy raw input space with the more compact semantic output space which has shown to improve recognition [26, 22].

To improve image classification with reduced training data, [4, 27] use attributes as an intermediate layer and incorporate unlabeled data, however, both works are in a classical semi-supervised learning setting similar to [5], while our setting is transfer learning. More specifically [27] propose to bootstrap classifiers by adding unlabeled data. The bootstrapping is constrained by attributes shared across classes. In contrast, we use attributes for transfer and exploit the similarity between instances of the novel classes. [4] automatically discover a discriminative attribute representation, while incorporating unlabeled data. This notion of attributes is different to ours as we want to use semantic attributes to enable transfer from other classes. Other directions to improve the quality of the intermediate representation include integrating metric learning [31, 16] or online methods [10] which we defer to future work.

## 3  Propagated Semantic Transfer (PST)

Our main objective is to robustly recognize novel categories by transferring knowledge from known classes and exploiting the similarity of the test instances. More specifically our novel approach called *Propagated Semantic Transfer* consists of the following four components: we employ semantic knowledge transfer from known classes to novel classes (Sec. 3.1); we combine the transferred predictions with labels for the novel classes (Sec. 3.2); a similarity metric is defined to achieve a robust graph structure (Sec. 3.3); we propagate this information within the novel classes (Sec. 3.4).

### 3.1  Semantic knowledge transfer

We first transfer knowledge using a semantic representation. This allows to include external knowledge sources. We model the relation between a set of $K$ known classes $y_1, \ldots, y_K$ to the set of $N$ novel classes $z_1, \ldots, z_N$. Both sets are disjoint, i.e. $\{y_1, \ldots, y_K\} \bigcap \{z_1, \ldots, z_N\} = \emptyset$. We use two strategies to achieve this transfer: i) an attribute representation that employs an intermediate representation of $a_1, \ldots, a_M$ attributes or ii) direct similarities calculated among the known object classes. Both work without any training examples for $z_n$, i.e. also for zero-shot recognition [11, 24].

**i) Attribute representation.**  We use the Direct-Attribute-Prediction (DAP) model [11], using our formulation [24]. An intermediate level of $M$ attribute classifiers $p(a_m|x)$ is trained on the known classes $y_k$ to estimate the presence of attribute $a_m$ in the instance $x$. The subsequent knowledge transfer requires an external knowledge source that provides class-attribute associations $a_m^{z_n} \in \{0, 1\}$ indicating if attribute $a_m$ is associated with class $z_n$. Options for such association information are discussed in Section 4.2. Given this information the probability of the novel classes $z_n$ to be present in the instance $x$ can then be estimated [24]:

$$p(z_n|x) \propto \prod_{m=1}^{M} \left(2p(a_m|x)\right)^{a_m^{z_n}}. \tag{1}$$

**ii) Direct similarity.**  As an alternative to attributes, we can use the $U$ most similar training classes $y_1, ..., y_U$ as a predictor for novel class $z_n$ given an instance $x$ [24]:

$$p(z_n|x) \propto \prod_{u=1}^{U} \left(2p(y_u|x)\right)^{y_u^{z_n}}, \tag{2}$$

where $y_u^{z_n}$ provides continuous normalized weights for the strength of the similarity between the novel class $z_n$ and the known class $y_u$ [24]. To comply with [23, 22] we slightly diverge from these models for the ImageNet and MPII Composites dataset by using a sum formulation instead of the probabilistic expression, i.e. for attributes $p(z_n|x) \propto \frac{\sum_{m=1}^{M} a_m^{z_n} p(a_m|x)}{\sum_{m=1}^{M} a_m^{z_n}}$, and for direct similarity $p(z_n|x) \propto \frac{\sum_{u=1}^{U} p(y_u|x)}{U}$. Note that in this case we do not obtain probability estimates, however, for label propagation the resulting scores are sufficient.

## 3.2 Combining transferred and ground truth labels

In the following we treat the multi-class problem as $N$ binary problems, where $N$ is the number of binary classes. For class $z_n$ the semantic knowledge transfer provides $p(z_n|x) \in [0, 1]$ for all instances $x$. We combine the best predictions per class, scaled to $[-1, 1]$, with labels $\hat{l}(z_n|x) \in \{-1, 1\}$ provided for some instances $x$ in the following way:

$$l(z_n|x) = \begin{cases} \gamma \hat{l}(z_n|x) & \text{if there is a label for } x \\ (1-\gamma)(2p(z_n|x)-1) & \text{if } p(z_n|x) \text{ is among top-}\delta \text{ fraction of predictions for } z_n \\ 0 & \text{otherwise.} \end{cases} \quad (3)$$

$\gamma$ provides a weighting between the true labels and the predicted labels. In the zero-shot case we only use predictions, i.e. $\gamma = 0$. The parameters $\delta, \gamma \in [0, 1]$ are chosen, similar to the remaining parameters, using cross-validation on the training set.

## 3.3 Similarity metric based on discriminative models for graph construction

We enhance transfer learning by exploiting also the neighborhood structure within novel classes, i.e. we assume a transductive setting. Graph-based semi-supervised learning incorporates this information by employing a graph structure over all instances. In this section we describe how to improve the graph structure as it has a strong influence on the final results [5]. The k-NN graph is usually built on the raw feature descriptors of the data. Distances are computed for each pair $(x_i, x_j)$ by

$$d(x_i, x_j) = \sum_{d=1}^{D} |x_{i,d} - x_{j,d}|, \quad (4)$$

where $D$ is the dimensionality of the raw feature space. We note that the visual representation used for label propagation can be independent of the visual representation used for transfer. While the visual representation for transfer is required to provide good generalization abilities in conjunction with the employed supervised learning strategy, the visual representation for label propagation should induce a good neighborhood structure. Therefore we propose to use the more compact output space trained on the known classes which we found to provide a much better structure, see Figure 5b. We thus compute the distances either on the M-dimensional vector of the attribute classifiers $p(a_m|x)$ with $M \ll D$, i.e.,

$$d(x_i, x_j) = \sum_{m=1}^{M} |p(a_m|x_i) - p(a_m|x_j)|, \quad (5)$$

or on the $K$-dimensional vector of object-classifiers $p(y_k|x)$ with $K \ll D$, i.e.

$$d(x_i, x_j) = \sum_{\kappa=1}^{K} |p(y_\kappa|x_i) - p(y_\kappa|x_j)|. \quad (6)$$

These distances are transformed into similarities with a RBF kernel: $s(x_i, x_j) = \exp\left(\frac{-d(x_i, x_j)}{2\sigma^2}\right)$. Finally, we construct a k-NN graph that is known for its good performance [15, 5], i.e.,

$$W_{ij} = \begin{cases} s(x_i, x_j) & \text{if } s(x_i, x_j) \text{ is among the k largest similarities of } x_i \\ 0 & \text{otherwise.} \end{cases} \quad (7)$$

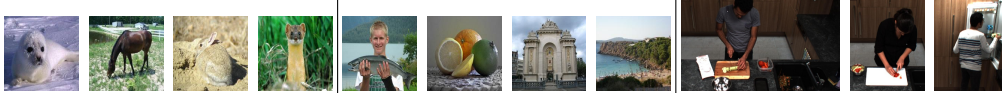

Figure 2: AwA (left), ImageNet (middle), and MPII Composite Activities (right)

### 3.4 Label propagation with certain and uncertain labels

In this work, we build upon the label propagation by [33]. The k-NN graph with RBF kernel gives the weighted graph $W$ (see Section 3.3). Based on this graph we compute a normalized graph Laplacian, i.e., $S = D^{-1/2}WD^{-1/2}$ with the diagonal matrix $D$ summing up the weights in each row in $W$. Traditional semi-supervised label propagation uses sparse ground truth labels. In contrast we have dense labels $l(z_n|x)$ which are a combination of uncertain predictions and certain labels (see Eq. 3) for all instances $\{x_1, \ldots, x_i\}$ of the novel classes $z_n$. Therefore, we modify the initialization by setting

$$L_n^{(0)} = [l(z_n|x_1), \ldots, l(z_n|x_i)] \tag{8}$$

for the $N$ novel classes. For each class, labels are propagated through this graph structure converging to the following closed form solution

$$L_n^* = (I - \alpha S)^{-1} L_n^{(0)} \quad \text{for } 1 \leq n \leq N, \tag{9}$$

with the regularization parameter $\alpha \in (0, 1]$. The resulting framework makes use of the manifold structure underlying the novel classes to regulate the predictions from transfer learning. In general, the algorithm converges after few iterations.

## 4 Evaluation

### 4.1 Datasets

We shortly outline the most important properties of the examined datasets in the following paragraphs and show example images/frames in Figure 2.

**AwA** The Animals with Attributes dataset (AwA) [11] is one of the first and most widely used datasets for semantic knowledge transfer and zero-shot recognition. It consists of $50$ mammal classes, $40$ training (24,395 images) and $10$ disjoint test classes (6,180 images). We use the provided pre-computed 6 image descriptors, which are concatenated.

**ImageNet** The ImageNet 2010 challenge [2] requires large scale and fine-grained recognition. It consists of 1000 image categories which are split into 800 training and 200 test categories according to [23]. We use the LLC and Fisher-Vector encoded SIFT descriptors provided by [23].

**MPII Composite Activities** The MPII Composite Cooking Activities dataset [22] distinguishes 41 basic cooking activities, such as *prepare scrambled egg* or *prepare carrots* with video recordings of varying length from 1 to 41 minutes. It consists of a total of 256 videos, 44 are used for training the attribute representation, 170 are used as test data. We use the provided dense-trajectory representation and train/test split.

### 4.2 External knowledge sources and similarity measures

Our approach incorporates external knowledge to enable semantic knowledge transfer from known classes $y$ to unseen classes $z$. We use the class-attribute associations $a_m^{z_n}$ for attribute-based transfer (Equation 1) or inter-class similarity $y_u^{z_n}$ for direct-similarity-based transfer (Equation 2) provided with the datasets. In the following we shortly outline the knowledge sources and measures.

**Manual (AwA)** AwA is accompanied with a set of 85 attributes and associations to all 40 training and all 10 test classes. The associations are provided by human judgments [11].

**Hierarchy (ImageNet)** For ImageNet the manually constructed WordNet/ImagNet hierarchy is used to find the most similar of the 800 known classes (leaf nodes in the hierarchy). Furthermore, the 370 *inner nodes* can group several classes into attributes [23].

| Approach | Performance | |
|---|---|---|
| | AUC | Acc. |
| DAP [11] | 81.4 | 41.4 |
| IAP [11] | 80.0 | 42.2 |
| Zero-Shot Learning [9] | n/a | 41.3 |
| PST (ours) | | |
|   on image descriptors | 81.2 | 40.5 |
|   on attributes | 83.7 | 42.7 |

(a) Zero-Shot. Predictions with attributes and manual defined associations, in %.

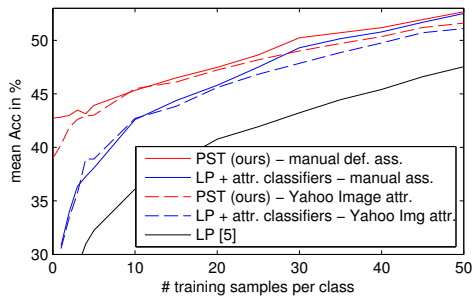

(b) Few-Shot

Figure 3: Results on AwA Dataset, see Sec. 4.3.1.

**Linguistic knowledge bases (AwA, ImageNet)** An alternative to manual association are automatically mined associations. We use the provided similarity matrices which are extracted using different linguistic similarity measures. They are either based on linguistic corpora, namely *Wikipedia* and *WordNet*, or on hit-count statistics of web search. One can distinguish basic web search (*Yahoo Web*), web search refined to part associations (*Yahoo Holonyms*), image search (*Yahoo Image* and *Flickr Image*), or use the information of the summary snippets returned by web search (*Yahoo Snippets*). As ImageNet does not provide attributes, we mined 811 part-attributes from the associated WordNet hierarchy [23].

**Script data (MPII Composites)** To associate composite cooking activities such as *preparing carrots* with attributes of fine-grained activities (e.g. *wash*, *peel*), ingredients (e.g. *carrots*), and tools (e.g. *knife*, *peeler*), textual description (*Script data*) of these activities were collected with AMT. The provided associations are computed based on either the *frequency statistics* or, more discriminate, by term frequency times inverse document frequency (*tf\*idf*). Words in the text can be matched to labels either *literally* or by using *WordNet* expansion [22].

## 4.3 Results

To enable a direct comparison, we closely follow the experimental setups of the respective datasets [11, 23, 22]. On all datasets we train attribute or object classifiers (for direct similarity) with one-vs-all SVMs using Mean Stochastic Gradient Descent [23] and, for AwA and MPII Composites, with a $\chi^2$ kernel approximation as in [22]. To get more distinctive representations for label propagation we train sigmoid functions [12] to estimate probabilities (on the training set for AwA/MPII Composites and on the validation set for ImageNet).

The hyper-parameters of our new *Propagated Semantic Transfer* algorithm are estimated using 5-fold cross-validation on the respective training set, splitting them into 80% known and 20% novel classes: We determine the parameters for our approach on the AwA training set and then set them for all datasets to $\alpha = 0.8$, $\gamma = 0.98$, the number of neighbors $k = 50$, the number of iterations for propagation to 10, and use $L1$ distance. Due to the different recognition precision of the datasets we determine $\delta = 0.15/0.04$ separately for AwA/ImageNet. For MPII Composites we only do zero-shot recognition and use all samples due to the limited number of samples of $\leq 7$ per class. For few-shot recognition we report the mean over 10 runs where we pick examples randomly. The labeled examples are included in the evaluation to make it comparable to the zero-shot case.

We validate our claim that the classifier output space induces a better neighborhood structure than the raw features by examining the k-Nearest-Neighbour (kNN) quality for both. In Figure 5b we compare the kNN quality on two datasets (see Sec. 4.1) for both feature representation. We observe that the attribute (Eq. 5) and object (Eq. 6) classifier-based representations (green and magenta dashed line) achieve a significantly higher accuracy than the respective raw feature-based representation (Eq. 4, Fig. 5b solid lines). We note that a good kNN-quality is required but not sufficient for good propagation, as it also depends on the distribution and quality of initial predictions. In the following, we compare the performance of the raw features with the attribute classifier representation.

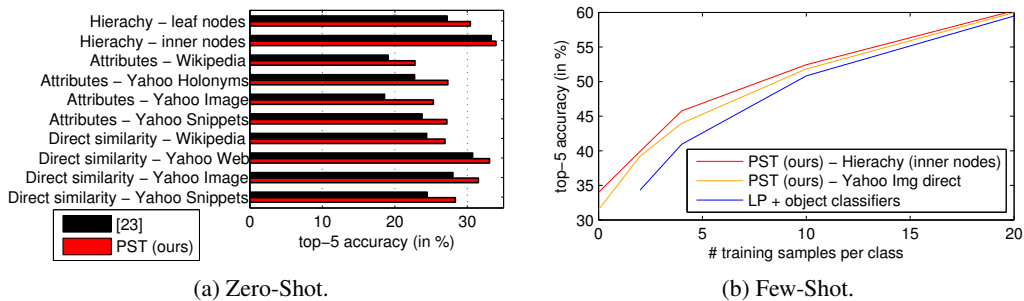

|                | (a) Zero-Shot. | (b) Few-Shot. |
|----------------|----------------|----------------|

Figure 4: Results on ImageNet, see Sec. 4.3.2.

### 4.3.1 AwA - image classification

We start by comparing the performance of related work to our approach on AwA (see Sec. 4.1) in Figure 3. We start by examining the zero-shot results in Figure 3a, where no training examples are available for the novel or in this case unseen classes. The best results to our knowledge for on this dataset are reported by [11]. On this 10-class zero-shot task they achieve 81.4% area under ROC-curve (AUC) and 41.4% multi-class accuracy (Acc) with DAP, averaged over the 10 test classes. Additionally we report results from Zero-Shot Learning [9] which achieves 41.3% Acc. Our *Propagated Semantic Transfer*, using the raw image descriptors to build a neighborhood structure, achieves 81.2% AUC and 40.5% Acc. However, when propagating on the 85-dimensional attribute space, we improve over [11] and [9] to 83.7% AUC and 42.7% Acc. To understand the difference in performance between the attribute and the image descriptor space we examine the neighborhood quality used for propagating labels shown in Figure 5b. The k-NN accuracy, measured on the ground truth labels, is significantly higher for the attribute space (green dashed curve) compared to the raw features (solid green). The information is more likely propagated to neighbors of the correct class for the attribute-space leading to a better final prediction. Another advantage is the significantly reduced computation and storage costs for building the k-NN graph which scales linearly with the dimensionality. We believe that such an intermediate space, in this case represented by attributes, might provide a better neighborhood structure and could be used in other label-propagation tasks.

Next we compare our approach in the few-shot setting, i.e. we add labeled examples per class. In Figure 3b we compare our approach (PST) to two label propagation (LP) baselines. We first note that PST (red curves) seamlessly moves from zero-shot to few-shot, while traditional LP (blue and black curves) needs at least one training example. We first examine the three solid lines. The black curve is our best LP variant from [5] evaluated on the 10 test classes of AwA rather than all 50 as in [5]. We also compute LP in combination with the similarity metric based on the attribute classifier scores (blue curves). This transfer of knowledge residing in the classifier trained on the known classes already gives a significant improvement in performance. Our approach (red curve) additionally transfers labels from the known classes and improves further. Especially for few labels our approach benefits from the transfer, e.g. for 5 labeled samples per class PST achieves 43.9% accuracy, compared to 38.1% for LP with attribute classifiers and 32.2% for [5]. For less samples LP drops significantly while our approach has nearly stable performance. For large amounts of training data, PST approaches - as expected - LP (red vs. blue in Figure 3b).

The dashed lines in Figure 3b provide results for automatically mined associations $a_m^{z_n}$ between attributes and classes. It is interesting to note that these automatically mined associations achieve performance very close to the manual defined associations (dashed vs. solid). In this plot we use Yahoo Image as base for the semantic relatedness, but we also provide the improvements of PST for the other linguistic language sources in supplemental material.

### 4.3.2 ImageNet - large scale image classification

In this section we evaluate our Propagated Semantic Transfer approach on a large image classification task with 200 unseen image categories using the setup as proposed by [23]. We report the top-5 accuracy[2] [2] which requires one of the best five predictions for an image to be correct.

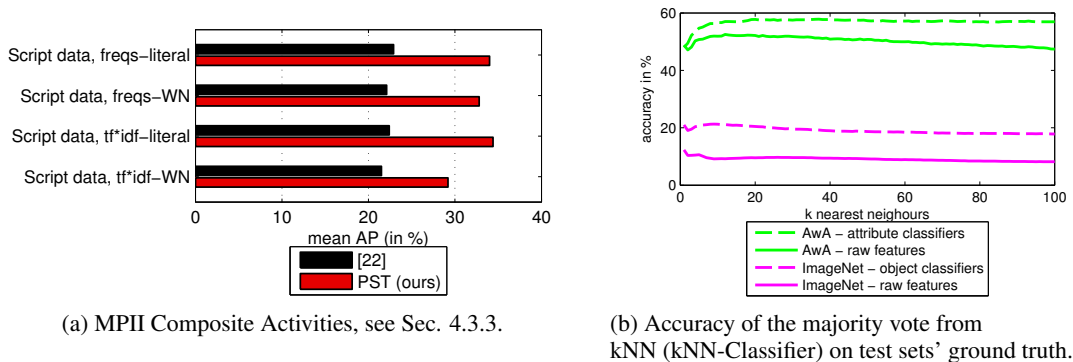

(a) MPII Composite Activities, see Sec. 4.3.3.

(b) Accuracy of the majority vote from kNN (kNN-Classifier) on test sets' ground truth.

Figure 5: Results

Results are reported in Figure 4. For zero-shot recognition our PST (red bars) improves performance over [23] (black bars) as shown in Figure 4a. The largest improvement in top-5 accuracy is achieved for Yahoo Image with Attributes which increases by 6.7% to 25.3%. The absolute performance of 34.0% top-5 accuracy is achieved by using the inner nodes of the WordNet hierarchy for transfer, closely followed by Yahoo Web with direct similarity, achieving 33.1% top-5 accuracy. Similar to the AwA dataset we improve PST over the LP-baseline for few-shot recognition (Figure 4b).

### 4.3.3 MPII composite - activity recognition

In the last two subsections, we showed the benefit of *Propagated Semantic Transfer* on two image classification challenges. We now evaluate our approach on the video-activity recognition dataset MPII Composite Cooking Activities [22]. We compute mean AP using the provided features and follow the setup of [22]. In Figure 5a we compare our performance (red bars) to the results of zero-shot recognition without propagation [22] (black bars) for four variants of Script data based transfer. Our approach achieves significant performance improvements in all four cases, increasing mean AP by 11.1%, 10.7%, 12.0%, and 7.7% to 34.0%, 32.8%, 34.4%, and 29.2%, respectively. This is especially impressive as it reaches the level of supervised training: for the same set of attributes (and very few, $\leq$ 7 training categories per class) [22] achieve 32.2% for SVM, 34.6% for NN-classification, and up to 36.2% for a combination of NN with script data.

We find these results encouraging as it is much more difficult to collect and label training examples for this domain than for image classification and the complexity and compositional nature of activities frequently requires recognizing unseen categories [9].

## 5 Conclusion

In this work we address a frequently occurring setting where there is large amount of training data for some classes, but other, e.g. novel classes, have no or only few labeled training samples. We propose a novel approach named *Propagated Semantic Transfer*, which integrates semantic knowledge transfer with the visual similarities of unlabeled instances within the novel classes. We adapt a semi-supervised label-propagation approach by building the neighborhood graph on expressive, low-dimensional semantic output space and by initializing it with predictions from knowledge transfer.

We evaluated this approach on three diverse datasets for image and video-activity recognition, consistently improving performance over the state-of-the-art for *zero-shot* and *few-shot* prediction. Most notably we achieve 83.7% AUC / 42.7% multi-class accuracy on the Animals with Attributes dataset for zero-shot recognition, scale to 200 unseen classes on ImageNet, and achieve up to 34.4% (+12.0%) mean AP on MPII Composite Activities which is on the level of supervised training on this dataset. We show that our approach consistently improves performance independent of factors such as (1) the specific datasets and descriptors, (2) different transfer approaches: direct vs. attributes, (3) types of transfer association: manually defined, linguistic knowledge bases, or script data, (4) domain: image and video activity recognition, or (5) model: probabilistic vs. sum formulation.

**Acknowledgements.** This work was partially funded by the DFG project SCHI989/2-2.

## Footnotes

[1]We use "novel" throughout the paper to denote categories with no or few labeled training instances.

[2]*top-5 accuracy* = 1 - *top-5 error* as defined in [2]

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
