[Supplementary Material]

# SUPPLEMENTAL MATERIAL
## of
# Transfer Learning in a Transductive Setting

**Marcus Rohrbach**　　　　**Sandra Ebert**　　　　**Bernt Schiele**

Max Planck Institute for Informatics, Saarbrücken, Germany
`{rohrbach,ebert,schiele}@mpi-inf.mpg.de`

## 1　Qualitative results

Figure 1 shows the graph structure build between the test images to transfer labels. As the test images are sorted by class, the block structure around the diagonal indicates that the graph structure can help to improve the labeling. This is also supported by our quantitative results.

## 2　Cross-validation parameter estimation

Figure 2 shows different $\gamma$, cross-validated on the training set of AwA. According to this figure we chose $\gamma = 0.98$ (red dashed line).

## 3　Result tables for additional semantic relatedness measures.

For detailed inspection we provide results for the alternative semantic knowledge sources.

- Table 1 shows results for the AwA dataset.
- Table 2 shows results for the ImageNet dataset and is the base of Figure 4a in the submission.
- Table 3 shows results for the MPII Composite dataset and is the base of Figure 5a in the submission.

We note that in all cases our PST approach outperforms related work and baselines.

Figure 1: Visualisation of kNN graph on test images for AwA dataset, manual associations.

(a) Accuracy

(b) AUC

Figure 2: 5-fold crossvalidation of parameter $\gamma$ on training set of AwA.

|                              | DAP [24] | DAP  | PST (ours) |
|------------------------------|----------|------|------------|
| Attributes - manual def. ass. | 78.5     | 80.4 | 83.7       |
| Attributes - WordNet         | 60.5     | 61.1 | 61.7       |
| Attributes - Wikipedia       | 69.7     | 70.4 | 77.0       |
| Attributes - Yahoo Web       | 60.4     | 61.1 | 62.0       |
| Attributes - Yahoo Image     | 71.0     | 74.3 | 81.1       |
| Attributes - Flickr Image    | 70.1     | 66.2 | 73.6       |
| Direct similarity - WordNet  | 73.4     | 73.9 | 76.4       |
| Direct similarity - Wikipedia | 76.6    | 78.9 | 80.9       |
| Direct similarity - Yahoo Web | 77.7    | 77.6 | 79.3       |
| Direct similarity - Yahoo Image | 78.8  | 78.7 | 80.0       |
| Direct similarity - Flickr Image | 77.8 | 78.6 | 80.2       |

Table 1: AwA dataset, mean AUC in %. The first column reports the results of [24], the second our reproduction, and the third results for our approach.

|                                  | [23] | PST (ours) |
|----------------------------------|------|------------|
| Hierachy - leaf nodes            | 27.2 | 30.4       |
| Hierachy - inner nodes           | 33.3 | 34.0       |
| Attributes - Wikipedia           | 19.1 | 22.8       |
| Attributes - Yahoo Holonyms      | 22.7 | 27.3       |
| Attributes - Yahoo Image         | 18.6 | 25.3       |
| Attributes - Yahoo Snippets      | 23.8 | 27.2       |
| Direct similarity - Wikipedia    | 24.4 | 26.9       |
| Direct similarity - Yahoo Web    | 30.7 | 33.1       |
| Direct similarity - Yahoo Image  | 28.0 | 31.5       |
| Direct similarity - Yahoo Snippets | 24.5 | 28.3     |

Table 2: ImageNet, top-5 accuracy in %. Comparision of our PST approach to results reported in [23], underlying table for bar plot in Figure 4a in our submission.

|                | [22] | PST (our) |
|----------------|------|-----------|
| freqs-literal  | 22.9 | 34.0      |
| freqs-WN       | 22.1 | 32.8      |
| tf*idf-literal | 22.4 | 34.4      |
| tf*idf-WN      | 21.5 | 29.2      |

Table 3: MPII Composite Activities Dataset, mean AP in %, underling table for bar plot in Figure 5a in our submission.