[Reviews · NeurIPS 2013]

Submitted by Assigned_Reviewer_6

This paper presents an approach to exploit the local similarity structure for zero-shot or a few-shot problem. The idea is to not only use mid-level representation such as attributes which help in zero-shot problem but also ensure that the unlabeled data is labeled such that similar images receive similar label.

Overall, I like the direction of the paper. I think exploiting graph structure is an interesting idea which hasn't been looked into the zero-shot problem (as far as I know). While there might be multiple approaches to incorporate that, this paper uses the label propagation technique. The experiments in the paper are quite convincing (performed on three different datasets) and I believe authors have incorporated some reasonable baselines.

However, I think paper can be improved..Here are some suggestions and issues:
1. The biggest issue with the paper is qualitative results. Can the authors show some labeling they get without using graph structure and w/ graph structure so that they can show how this helps.
2. I think figure 1 is also not quite easy to understand. Until you read the details of what y and z are it is unclear what to get out of figure 1. I think it can easily be made more accessible with examples.
3. The paper should cite some recent work in SSL which also looks into graph structure based on attributes etc:
Abhinav Shrivastava, Saurabh Singh, Abhinav Gupta.Constrained Semi-Supervised Learning Using Attributes and Comparative Attributes. In ECCV 2012.
Another similar paper to cite is:
Adding Unlabeled Samples to Categories by Learned Attributes
Jonghyun Choi, Mohammad Rastegari, Ali Farhadi, Larry Davis, CVPR 2013
Summary: Overall I like the paper. The direction seems intuitive and paper does reasonable set of experiments to convince about the idea.

Submitted by Assigned_Reviewer_7

This paper describes how to attack the zero-, one-, or few-shot recognition problem, where we have a fair amount of training data for some classes, but none or very few for some other classes. It does this using three different techniques, all combined in a single framework: using semantically-meaningful mid-layer knowledge (attributes), building a graph on new classes to exploit the manifold structure, and finally by using an attribute-based representation for building the graph structure (rather than low-level features), which improves performance. The method is evaluated on 3 different datasets (Animals with Attributes, ImageNet, and MPII Cooking composites), and shows improved performance on all compared to the state-of-the-art (slightly).

On the plus side, the paper's methods are well-motivated and can fit into the same optimization framework, regardless of which kinds of side- or ground truth-annotations are available. The experiments appear to be conducted fairly and show slight improvements on the existing state-of-the-art (significantly for the MPII dataset). The paper is well-written.

The main negative is simply that it is somewhat surprising that performance is not improved more by doing all of this. I also have some minor suggestions to make the paper a bit easier to read.

- Exploiting the similarities of new classes w.r.t. existing classes seems very similar to the "simile classifiers" of Kumar et al. in their ICCV 2009 paper "Face verification using attribute and simile classifiers."

- line 74: rewrite to "...models do not have to be retrained..."

- Figure 3b: reorder legend to put red curves on top, then blue curves, then black

- Figure 4a: accuracy is misspelled

- supplementary material table 2 caption: underlying is misspelled
Summary: Good paper bringing together multiple ideas to do few-shot recognition.

Submitted by Assigned_Reviewer_8

This paper builds upon previous transfer learning (zero-shot learning) work [24,23] wherein the idea of "semantic relatedness" was introduced. It augments the previous work by introducing the idea of transductive learning from [33]. By using the visual similarities of unlabeled and labeled samples, a graph structure is built and used to propagate labels to novel classes. Unlike previous work [33], they improve the graph structure by replacing noisy raw input space with more informative attribute space.

Paper is well-written and easy to read. Proposed idea has been validated using relevant experiments. Results do show the benefit of introducing transductive learning into transfer learning framework.

Summary: Paper nicely builds upon two previous works i.e., semantic-relatedness transfer learning [24] and label-propagation [33]. Idea introduced is interesting.
Author Feedback

Author rebuttal: We would like to thank the reviewers for their constructive feedback. All reviewers acknowledge our relevant/convincing experiments which improve performance over state-of-the-art. The reviewers recognize that we present a novel, interesting idea [R6], and use a well-motivated method [R7] which is composed in a well written paper [R7,R8].



R6:
We agree that Figure 1 is suboptimal and that qualitative results would be beneficial. Following these suggestions we are happy to add qualitative results which will also help to illustrate Figure 1 more clearly. More extensive results will be provided as supplemental material.

Additionally R6 suggests citing Shrivastava et al. [ECCV 12] and the very recent work of Choi et al. [CVPR 13]. We will discuss both papers as part of the related work section in the final version.
Similar to us, both use attributes as an intermediate layer and incorporate unlabeled data to improve image classification with reduced training data. Both works are in a classical SSL setting similar to [4], while our setting is transfer learning. This means we transfer knowledge from categories where sufficient training data is available, which allows to recognize novel categories with fewer or no training samples.
More specifically Shrivastava et al. propose to bootstrap classifiers by adding unlabeled data. The bootstrapping is constrained by attributes shared across classes. In contrast, we use attributes for transfer and exploit the similarity between instances of the novel classes.
Choi et al. automatically discover a discriminative attribute representation, while incorporating unlabeled data. This notion of attributes is different to ours as we want to use semantic attributes to enable transfer from other classes.



R7:
R7 notes that “exploiting the similarities of new classes w.r.t. existing classes seems very similar to the simile classifiers” proposed by Kumar et al. [ICCV 09]. The “simile classifiers” are similar to the direct similarity approach from [24], Equation (2). The difference is that they focus on a localized part on the image, useful in face recognition. The “simile classifiers” could replace direct similarity or attribute classifiers in our approach if this information is available for a given dataset. Although this is not the focus of this work, we can discuss this work if the reviewers and area chairs wish.

We like to thank R7 for the detailed comments with respect to formulation and clarity. We will incorporate them in the final version.



R8:
Thanks to R8 for noting that the paper title is incorrect in CMT, we will correct this.